# Time-Optimal Velocity Tracking Control for Consensus Formation of Multiple Nonholonomic Mobile Robots

**DOI:** 10.3390/s21237997

**Published:** 2021-11-30

**Authors:** Hamidreza Fahham, Abolfazl Zaraki, Gareth Tucker, Mark W. Spong

**Affiliations:** 1Institute of Railway Research, School of Computing and Engineering, University of Huddersfield, Huddersfield HD1 3DH, UK; G.J.Tucker@hud.ac.uk; 2Centre for Artificial Intelligence, Robotics and Human-Machine Systems (IROHMS), Cardiff University, Cardiff CF24 3AA, UK; zarakia@cardiff.ac.uk; 3Erik Jonsson School of Engineering & Computer Science, University of Texas at Dallas, 800 W Campbell Rd., Richardson, TX 75080, USA; mspong@utdallas.edu

**Keywords:** time-optimal, velocity tracking, consensus formation, switching control, multi-robot systems

## Abstract

The problem of velocity tracking is considered essential in the consensus of multi-wheeled mobile robot systems to minimise the total operating time and enhance the system’s energy efficiency. This study presents a novel switched-system approach, consisting of bang-bang control and consensus formation algorithms, to address the problem of time-optimal velocity tracking of multiple wheeled mobile robots with nonholonomic constraints. This effort aims to achieve the desired velocity formation in the least time for any initial velocity conditions in a multiple mobile robot system. The main findings of this study are as follows: (i) by deriving the equation of motion along the specified path, the motor’s extremal conditions for a time-optimal trajectory are introduced; (ii) utilising a general consensus formation algorithm, the desired velocity formation is achieved; (iii) applying the Pontryagin Maximum Principle, the new switching formation matrix of weights is obtained. Using this new switching matrix of weights guarantees that at least one of the system’s motors, of either the followers or the leader, reaches its maximum or minimum value by using extremals, which enables the multi-robot system to reach the velocity formation in the least time. The proposed approach is verified in a theoretical analysis along with the numerical simulation process. The simulation results demonstrated that using the proposed switched system, the time-optimal consensus algorithm behaved very well in the networks with different numbers of robots and different topology conditions. The required time for the consensus formation is dramatically reduced, which is very promising. The findings of this work could be extended to and beneficial for any multi-wheeled mobile robot system.

## 1. Introduction

Thanks to the greater functionality and performance of cooperative, mobile robot systems versus single mobile robots, they are highly beneficial and used in many applications such as hazardous material handling, surveillance, environment exploration, transportation of large objects, etc. Although the cooperative nature of these systems may result in greater efficiency and operational capability compared to a single mobile robot, it introduces a challenging control problem that must deal with consensus formation. For instance, in the works presented in [1,2] there are two examples in the domain of formations control for multi-agent systems in surveillance applications, and the control of spacecraft using formations control is presented in [3,4,5,6].

The study of consensus formation control of multi-agent systems (Figure 1) can be classified into three categories: a leader–follower approach [7,8,9], the virtual structure approach [10,11], and decentralised control [12,13]. The leader–follower approach consists of a leader agent, a robot/vehicle or human operator who is entrusted to track a specified trajectory, and the follower agents are designed to follow the leader agent while achieving the desired formation and heading consensus. This approach is very useful, especially in surveillance and exploration applications. The virtual structure approach considers a multi-agent system as a single rigid robot with multiple input controllers. This approach is useful for large object transportation. The advantage of such an algorithm is the simplicity of the controller design. The decentralised approach, which includes the nearest neighbour approach, allows each robot to decide based on interactions with its neighbours. The advantage of this approach is its robustness against disturbances which makes it more suitable for military manoeuvres or automated highway applications [14,15,16,17,18]. Das et al. in [14] and Cao et al. in [15] nicely reviewed the cooperative search and formation control methods for multiple autonomous vehicles, which are helpful to learn more about these methods. Furthermore, a tutorial overview, together with several specific applications of consensus algorithms, was provided in [16]. Ren et al. in [17] addressed the double integrator dynamics of the agents in consensus algorithms and discussed the theoretical and practical aspects of this topic. Zhang et al. in [13] addressed a two-stage cooperative guidance strategy to achieve a salvo attack in directed topologies of multiple interceptors. At the first stage, a prescribed-time optimal consensus method was constructed offline. A novel fixed-time distributed guidance law based on the proportional navigation guidance law was designed by integrating a consistent control technique into the guidance strategy [19]. As shown in [13,19], the optimal consensus strategy was used along with other navigation systems for the control of multi-systems, such as for missiles.

This paper considers the time-optimal velocity tracking problem of multiple wheeled mobile robots, extending the single robot case treated in [18]. Despite many previous works on the optimal control of multi-agent systems [20,21,22,23,24,25,26,27], we strongly believe that the methodologies and objectives of our work are novel as they differ from the state-of-the-art. To get a better comparison, the previous representative works in this domain are listed in Table 1.

To the best of our knowledge, no research has been reported in which the motion time consensus along the desired formation of a group of mobile robots, to form and maintain a desired geometric pattern and follow the desired trajectory, is addressed as the objective function using a switching mechanism based on bang-bang control. In short, the main goal of this work is to find control inputs of a leader–follower multi-robot system to achieve the desired consensus velocity tracking formation in the least time.

## 2. Dynamics of Wheeled Mobile Robots

This section presents the dynamics of the wheeled mobile robots (WMR) and the consensus formation of a multi-agent system.

A schematic of a WMR is shown in Figure 2. Following [18], the kinematic model of the *i*th WMR with two degrees of freedom can be written as follows
(1)x˙i=vicos(θi),   y˙i=vi sin(θi),   θ˙i=ωi
where, vi∈ℝ and ωi∈ℝ denote the forward and angular velocity of the *i*th WMR, respectively.

The configuration of the robot is given by (xi, yi,θi)∈ℝ3, where (xi, yi) is the Cartesian position of the centre of mass of the robot and θi is the heading angle of the robot with respect to the X axis of the fixed coordinate system.

The assumption that the wheels do not slip gives rise to the nonholonomic constraint (2)
(2)$${\dot{x}}_{i}\sin\left(\theta_{i}\right)-{\dot{y}}_{i}\cos\left(\theta_{i}\right)=0$$

In the Equations (3)–(6), let ${\dot{\phi }}_{R_{i}}$, ${\dot{\phi }}_{RL_{i}}$, *2b*, and *r* be the right wheel angular velocity, the left wheel angular velocity, the distance between the wheels and the radii of the wheels, respectively. The relation between the forward and angular velocities to the wheel velocities can then be presented as follows
(3)$$v_{i}+b\omega_{i}=r\;{\dot{\phi }}_{R_{i}}$$
(4)$$v_{i}-b\omega_{i}=r\;{\dot{\phi }}_{L_{i}}$$
(5)$$v_{i}=\frac{r}{2}\;{\dot{\phi }}_{R_{i}}+\frac{r}{2}\;{\dot{\phi }}_{L_{i}}$$
(6)$$\omega_{i}=\frac{r}{2b}\;{\dot{\phi }}_{R_{i}}-\frac{r}{2b}\;{\dot{\phi }}_{L_{i}}$$

Next, the net torque $\tau_{R_{i}}$ and $\tau_{L_{i}}$ need to be applied by motors at the right and left wheels, respectively. By using Newton-Euler formulations, the dynamics equations of motion of the *i*th WMR are given by
(7)$$m_{i}{\dot{v}}_{i}=\frac{1}{r}\;\tau_{R_{i}}+\frac{1}{r}\;\tau_{L_{i}}$$
(8)$$J_{i}{\dot{\omega }}_{i}=\frac{b}{r}\;\tau_{R_{i}}-\frac{b}{r}\;\tau_{L_{i}}$$
where $m_{i}$ and $J_{i}$ are the mass and the effective rotational inertia about the vertical axis through the center of mass of the *i*th WMR, respectively. Thus, the Equations (7) and (8) can be rewritten in matrix form as
(9)$$\left[\begin{array}{c} {\dot{v}}_{i}\\ {\dot{\omega }}_{i} \end{array}\right]=\left[\begin{array}{cc} \frac{1}{m_{i}\;r} & \frac{1}{m_{i\;}r}\\ \frac{b}{J_{i}\;r} & -\frac{b}{J_{i}\;r} \end{array}\right]\left[\begin{array}{c} \tau_{R_{i}}\\ \tau_{L_{i}} \end{array}\right]$$

Using Equations (5)–(8) the right and left wheel angular accelerations are
(10)$$\left[\begin{array}{c} \tau_{R_{i}}\\ \tau_{L_{i}} \end{array}\right]=C{\ddot{\Phi }}_{i}=\left[\begin{array}{cc} c_{1} & c_{2}\\ c_{2} & c_{1} \end{array}\right]\left[\begin{array}{c} {\ddot{\phi }}_{R_{i}}\\ {\ddot{\phi }}_{L_{i}} \end{array}\right]$$
where $c_{1}=\frac{r^{2}}{4}\;m_{i}+\frac{r^{2}}{4b^{2}}\;J_{i}$ and $c_{2}=\frac{r^{2}}{4}\;m_{i}-\frac{r^{2}}{4b^{2}}\;J_{i}$ are constant which depend on the mass and mass distribution of the WMR. The coefficient matrix in Equation (10) is easily seen to be nonsingular.

For a system of N robots, suppose the dynamics of the robots are described by:(11)$${\dot{V}}_{i}=M^{-1}u_{i}$$
where ${\dot{V}}_{i}={\left[\begin{array}{cc} {\dot{v}}_{i} & {\dot{\omega }}_{i} \end{array}\right]}^{T}$ and $u_{i}={\left[\begin{array}{cc} \tau_{R_{i}} & \tau_{L_{i}} \end{array}\right]}^{T}$ and *M* is the inertia matrix. Considering Equations (9) and (10), we can obtain Equation (12) as follows
(12)$${\ddot{\Phi }}_{i}=C^{-1}M{\dot{V}}_{i}$$

## 3. Time-Optimal Control of WMRs

In the time-optimal problem, the cost function is defined initially as
(13)$$J=t_{f}=\int \frac{dq}{\dot{q}}$$

According to the Pontryagin Maximum Principle, at each time of the time-optimal trajectory, at least one of the inputs reaches its extremal [33,34]. Let the inputs $\tau_{R}$ and/or $\tau_{L}$ have the maximum value $\tau_{M}$. Thus, the bounds of the torques are:(14)$$-\tau_{M}\leq \tau_{R_{i}}\leq \tau_{M},-\tau_{M}\leq \tau_{L_{i}}\leq \tau_{M}$$

By substituting the equations of motion along a specified path (12) into the constraint (14), the limitations on the torques are transferred to the limitations on acceleration as:(15)$$\frac{-c_{1}\tau_{M}-c_{2}\tau_{L}}{c_{1}^{2}-c_{2}^{2}}\leq {\ddot{\phi }}_{R_{i}}\leq \frac{c_{1}\tau_{M}-c_{2}\tau_{L}}{c_{1}^{2}-c_{2}^{2}}$$
(16)$$\frac{-c_{2}\tau_{R}-c_{1}\tau_{M}}{c_{1}^{2}-c_{2}^{2}}\leq {\ddot{\phi }}_{L_{i}}\leq \frac{-c_{2}\tau_{R}+c_{1}\tau_{M}}{c_{1}^{2}-c_{2}^{2}}$$

Therefore, there are four boundary angular accelerations of the wheel, and so there are four time-optimal control inputs when only one of the inputs exert its extremal.
$$\mathrm{EXT}1:\;\tau_{R}=\tau_{M}\;\Rightarrow {\ddot{\phi }}_{R_{i}}=\frac{c_{1}\tau_{M}-c_{2}\tau_{L}}{c_{1}^{2}-c_{2}^{2}}$$
$$\mathrm{EXT}2:\;\tau_{R}=-\tau_{M\;}\;\Rightarrow {\ddot{\phi }}_{R_{i}}=\frac{-c_{1}\tau_{M}-c_{2}\tau_{L}}{c_{1}^{2}-c_{2}^{2}}$$
$$\mathrm{EXT}3:\;\tau_{L}=\tau_{M\;}\;\Rightarrow {\ddot{\phi }}_{L_{i}}=\frac{-c_{2}\tau_{R}+c_{1}\tau_{M}}{c_{1}^{2}-c_{2}^{2}}$$
$$\mathrm{EXT}4:\;\tau_{L}=-\tau_{M\;}\;\Rightarrow {\ddot{\phi }}_{L_{i}}=\frac{-c_{2}\tau_{R}-c_{1}\tau_{M}}{c_{1}^{2}-c_{2}^{2}}$$

Moreover, it is possible that both motors are in extremals at the same time. By considering Equation (10), we see that these conditions will occur when the terms J become zero or $\tau_{R}{}_{i}=\tau_{R}{}_{i}=\tau_{M\;}$. This means that (i) the path does not have any curvature during this time, and therefore the WMR is moving in a straight line, or (ii) the path is irregular and the centre of mass stops at a stationary point and the robot is just rotating counterclockwise (CCW) or clockwise (CW) about the vertical axis. Therefore, the four additional extremal conditions are as follows:$$\mathrm{EXT}5:\;{\ddot{\phi }}_{R_{i}}=\frac{c_{1}\tau_{M}-c_{2}\tau_{M}}{c_{1}^{2}-c_{2}^{2}}\;\;\;\mathrm{and}\;{\ddot{\phi }}_{L_{i}}=\frac{c_{1}\tau_{M}-c_{2}\tau_{M}}{c_{1}^{2}-c_{2}^{2}}$$
the WMR is moving along a straight line and accelerating.
$$\mathrm{EXT}6:\;{\ddot{\phi }}_{R_{i}}=\frac{-c_{1}\tau_{M}-c_{2}\tau_{M}}{c_{1}^{2}-c_{2}^{2}}\;\;\;\mathrm{and}\;{\ddot{\phi }}_{L_{i}}=\frac{-c_{1}\tau_{M}-c_{2}\tau_{M}}{c_{1}^{2}-c_{2}^{2}}$$
the WMR is moving along a straight line and decelerating.
$$\mathrm{EXT}7:\;{\ddot{\phi }}_{R_{i}}=\frac{c_{1}\tau_{M}-c_{2}\tau_{M}}{c_{1}^{2}-c_{2}^{2}}\;\;\;\mathrm{and}\;{\ddot{\phi }}_{L_{i}}=\frac{-c_{1}\tau_{M}-c_{2}\tau_{M}}{c_{1}^{2}-c_{2}^{2}}$$
the WMR is rotating CCW about its centre of mass.
$$\mathrm{EXT}8:\;{\ddot{\phi }}_{R_{i}}=\frac{-c_{1}\tau_{M}-c_{2}\tau_{M}}{c_{1}^{2}-c_{2}^{2}}\;\mathrm{and}\;{\ddot{\phi }}_{L_{i}}=\frac{c_{1}\tau_{M}-c_{2}\tau_{M}}{c_{1}^{2}-c_{2}^{2}}$$
the WMR is rotating CW about its centre of mass.

## 4. Time-Optimal Consensus Algorithm Strategies

This section presents a time-optimal consensus algorithm that is designed and implemeted to reduce the convergence time in a multi mobile two-wheeled robot system. The consensus algorithm is used to impose similar dynamics on the information state of each robot in a group of robots. Every robot updates the value of its information state based on the information states of its neighbours. The goal of the consensus algorithm is to design a control law so that the information states of all of the robots converge to a similar value.

The communication among agents can be represented by directed or undirected graphs. A directed graph consists of a formation control graph *G* = (*N*, *E*, *D*), where *N* = {1, 2,… *i*, *j*, …, *n*} is a finite set of *n* vertices and a map dedicates to each vertex, a control system ${\dot{x}}_{i}=f_{i}\left(t,x_{i},u_{i}\right)$, a set of corresponding edges $E=\left\{e_{ij}≜\left(i,j\right)\in N\times N\right\}$, and a collection D={dij} of each edge $j:\;e_{ij}\in E$ that defines its control objective for some i∈N.

The edge eij∈E denotes the agent *j* can obtain information from agent *i*, but not necessarily vice versa. In undirected graphs, robots *i* and *j* are defined as neighbours of each other if there is an edge between them. In a system with *n* agents, suppose the dynamics of the agents are given by
(17)q˙i=C−1ui ,   i=1, 2, …, n,    qi, ui ∈ℝn
where q˙i denotes the acceleration of the *i*th agent and the ui is its input. The formation is described by
(18)D={dij:=qid−qjd}
where qid and qjd are the desired velocity states of agents *i* and *j*, who are neighbours of each other. The n×n adjacency matrix A=[aij] of a weighted graph is defined as aii=0 and aij=1 if i≠j, (i, j)∈E. Laplacian is defined as L=[lij] where lii=∑j≠iaij and lij=−aij if i≠j. Thus the Laplacian matrix is
(19)L=D−A

Suppose the robot *r* is designated as the leader and the other robots are the followers. We assume that a robot leader has direct access to the reference velocities ($v_{r},\;\omega_{r}$). Other robots (followers) use their neighbouring leader information to accomplish the formation tracking task. Regarding the consensus algorithm, the velocity tracking control for robot *i* is
(20)$$\{\begin{array}{c} q_{i}={\left[\begin{array}{cc} {\dot{\phi }}_{R_{i}} & {\dot{\phi }}_{L_{i}} \end{array}\right]}^{T}\\ \begin{array}{c} {\dot{q}}_{i}={\dot{q}}_{r}-\alpha_{i}l_{ii}\left(q_{i}-q_{r}\right)-\underset{j\in E}{\sum }l_{ij}(q_{i}-q_{j})\\ {\dot{P}}_{e}=-K_{EXT}\left(L⊗I_{2}\right)P_{e} \end{array}\\ p_{e_{i}}=q_{i}-q_{r} \end{array}$$
where $K_{EXT}=\alpha I_{2n}$ is the positive coefficient matrix so that
$$\alpha =\frac{\tau_{M}}{\mathrm{Max}\left\{\left|\tau_{R_{1}}\right|\;,\left|\tau_{L_{1}}\right|\;,\left|\tau_{R_{2}}\right|\;,\left|\tau_{L_{2}}\right|,\;\ldots ,\;\left|\tau_{R_{n}}\right|\;,\left|\tau_{L_{n}}\right|\;\right\}}$$

The whole algorithm is shown in Figure 3. Using the diagonal coefficient matrix $K_{EXT}$ in Equation (16), at least one of the right or left motors of the follower robots works at a maximum angular acceleration or deceleration. Therefore, the bang-bang controls take their extremals throughout the whole motion to minimise the manoeuvre time.

Following the α calculation step, the optimal control problem is now to find a control strategy to transfer the system state from the initial to the desired final state, in the least time (Figure 4). Since the Pontryagin principle demonstrates that at any point of a time-optimal trajectory, at least one of the actuators exerts a maximum or minimum input, the problem of finding the optimal control is reduced to finding the switching points. Therefore, the trajectory must be at each point tangent to one of the eight extremals described in Section 3, at least for one of the robots. Figure 5 shows an example of using the proposed algorithm. As can be seen, there are several switching points that occurred between actuators and extremals.

Having $\dot{q}$ versus *q*, the minimum time can be obtained using Equation (13). Then, from Equations (3)–(6) and (10), the angular accelerations and velocities of the wheels are obtained versus the time for the optimum trajectory.

**Theorem** **1.***The time-optimal consensus algorithm (20) solves the formation velocity tracking problem if the formation graph G has a spanning tree*.

**Proof.** As shown in [16,35,36], when the network topology of the multi-robot system contains a directed spanning tree, the rank of the Laplacian matrix L is *n*–1, zero is a simple eigenvalue of L with associated vector $b={\left[1\;\cdots 1\right]}^{-1}$ and all other eigenvalues are positive, and the consensus is reached asymptotically for the system
$${\dot{P}}_{e}=-\left(L⊗I_{2}\right)P_{e}.\;\mathrm{It}\;\mathrm{follows}\;\mathrm{that}\;\underset{t\to \infty }{\lim}e^{-\left(L⊗I_{2}\right)t}\to 1P_{e}\left(0\right)=0.$$In this proposed algorithm $K_{EXT}$ is a diagonal matrix in which all the diagonal elements are positive and also equal to or bigger than 1 (α≥1). Thus, it is a positive definite matrix, and the system asymptotically converges to the desired velocity configuration. Because of the time-optimal matrix coefficient component
$$\alpha =\frac{\tau_{M}}{\mathrm{Max}\left\{\left|\tau_{R_{1}}\right|\;,\left|\tau_{L_{1}}\right|\;,\left|\tau_{R_{2}}\right|\;,\left|\tau_{L_{2}}\right|,\;\ldots ,\;\left|\tau_{R_{n}}\right|\;,\left|\tau_{L_{n}}\right|\;\right\}}\geq 1$$Using the $K_{EXT}$ matrix instead of the identity matrix in the consensus algorithm (20) guarantees that at least one of the actuators of either the leader or followers works on the extremum conditions (EXT1 to EXT8 described in Section 3), the $\underset{t\to \infty }{\lim}e^{-eK_{EXT}\left(L⊗I_{2}\right)t}$ goes to zero faster, and the system asymptotically converges to the desired velocity configuration in the shortest time. □

## 5. Simulations

### 5.1. Comparison of the Performance of Velocity Tracking Consensus Formation with and without a Time-Optimal Strategy

In order to validate the proposed time-optimal algorithm, a computer program in Matlab software was developed to simulate two examples. All the conditions in these two examples are considered to be similar, including the initial conditions, the dimensions of the robots, the geometry of the desired path, and the topology of the robots’ network (Figure 5).

In the first simulation, where the time-optimal control strategy is not considered, the consensus velocity tracking algorithm is used whereas the time-optimal coefficient α is equal to 1, and the diagonal matrix $K_{EXT}$ is chosen as the identity matrix. As can be seen in Figure 6, the angular velocities and angular accelerations reach a consensus in 1.5 s.

In the second simulation, where the time-optimal control strategy is considered, the result of angular velocities and angular accelerations are shown in Figure 7. As can be seen, the proposed algorithm, based on consensus and bang-bang control, drives the motor torques to extremum values several times to achieve time-optimal control. According to Figure 7, the angular velocities and angular accelerations reach consensus in 0.4 s.

In order to compare the results of both simulations, Figure 8 shows the performance of the time-optimal consensus velocity tracking algorithm. The dashed lines show the normal consensus algorithm results, and the solid lines show the results of the proposed time-optimal consensus algorithm.

To make the performance of our simulations closer to the performance in the real-world setting, we considered adding some random disturbances into the systems. Figure 9 shows the results of the proposed time-optimal consensus algorithm in the presence of some random noises. The results show great stability of the consensus process on the discontinuity of switched controllers in the presence of noise. When the noise exists, there is less than a 10% error for remaining in angular acceleration and less than a 2% error in the angular velocity; however, the convergence is fine.

### 5.2. The Effect of Network Topology and the Number of Agents on the Convergence Time

In order to validate the proposed method in much more complex situations, the proposed method is implemented for a different number of robots. Figure 10 shows the effect of the number of robots on the convergence time of the velocity tracking control consensus formation with and without the time-optimal algorithm. Following the state-of-the-art findings in this field, we learned that the convergence time depends on several factors, such as the network topology, initial condition, formation control, number of robots, etc. Therefore, to explore the effect of the number of robots on the final convergence time, the initial conditions of all agents are considered to be similar in every simulation trial.

In every simulation trial, the robot network is considered fully interconnected, which means that every agent can send and receive information to all the other agents. Our findings demonstrated that the convergence time is reduced by increasing the number of agents. One reason for this could be the dimension and values of the gain matrix elements, which increase when the number of robots increases. As shown in Figure 10 and Figure 11 (right column), the time-optimal velocity tracking consensus formation converged faster, in all those examples, than the standard consensus formation without applying the time-optimal algorithm (Figure 10 and Figure 11, left column).

The results of the previous trial (Figure 10) show an increase in the convergence time when the number of robots increases. In order to investigate further the relationship between the graph connectivity and the convergence speed of consensus algorithms, a second trial was implemented. In this trial, the effect of the robot’s network topology and interconnectivity on the convergence time was investigated. Figure 11 compares the results of the consensus formation of two different topologies. Figure 11a shows a robot network with a fully interconnected structure, while Figure 11b shows a partially interconnected robot network where two connections are missing. As shown, the connections between robot number 6 and the robots numbered 4 and 5 are missing. Comparing these two topologies shows that the convergence time will increase when the network is not fully interconnected. As Figure 11c shows, the connections between four pairs of robots are missing. Although the proposed method managed to handle the convergence, it demonstrated an oscillatory behaviour in the angular velocity of the wheels. In other words, the developed bang-bang algorithm provides better support and convergence when the topology is fully connected. It tends to give more oscillatory results when the topology is not fully interconnected.

Comparing the results achieved in this section (shown in Figure 10 and Figure 11), it can be concluded that the convergence time is mainly influenced by two factors i.e., the number of robots in the network and also the network topology. However, as examined, the proposed time-optimal method is efficient if used when both of these factors are changed and it dramatically reduces the convergence time.

## 6. Conclusions

In this paper, an optimal time velocity tracking of consensus formation algorithm for multiple nonholonomic wheeled robots is proposed. To apply the motors’ extremum conditions for finding the time-optimal trajectory, firstly, the equation of motion along a specified path is obtained versus the left- and right-hand motors’ angular acceleration and angular velocities. Then by developing a hybrid Pontryagin principle of time-optimal control and a consensus algorithm for the multi-agent system, the novel switched system algorithm is introduced, and the time-optimal control problem is solved. By using the proposed method, the system achieved the desired velocity tracking formation in the optimal time. The simulation results of this novel switched-system approach and a typical consensus formation algorithm are provided and compared to demonstrate the effectiveness of the proposed approach. In addition to testing the performance of the proposed method, the effect of the robot network topology as well as the number of robots in a network are explored. The numerical simulation results demonstrated that the time-optimal consensus algorithm behaved very well and reduced remarkably the consensus formation time.

## Figures and Tables

**Figure 1 sensors-21-07997-f001:**
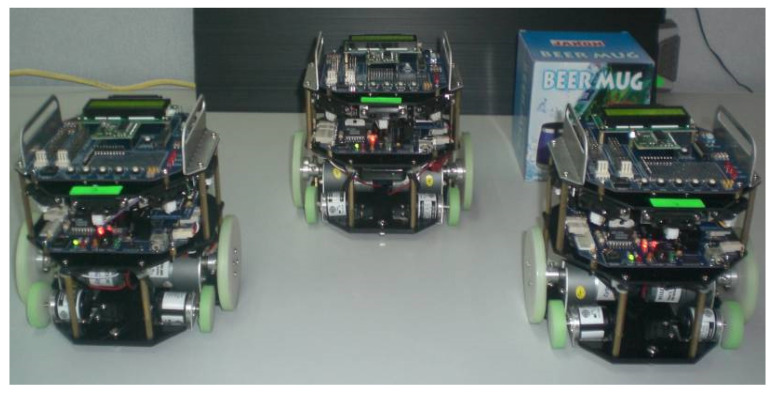
A group of nonholonomic wheeled mobile robots.

**Figure 2 sensors-21-07997-f002:**
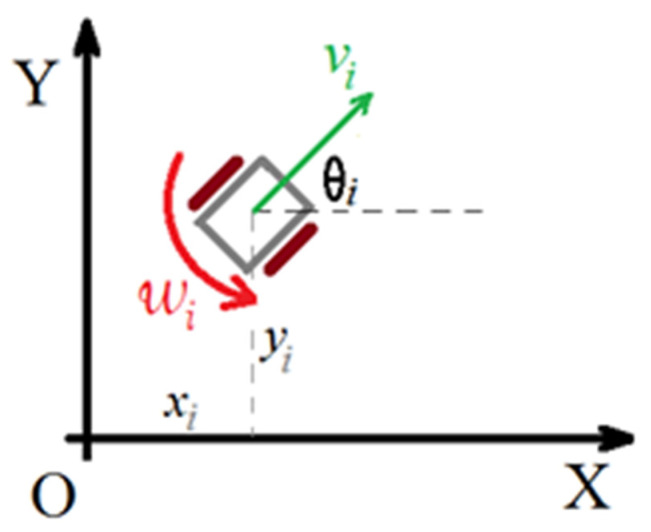
Schematic of a wheeled mobile robot.

**Figure 3 sensors-21-07997-f003:**
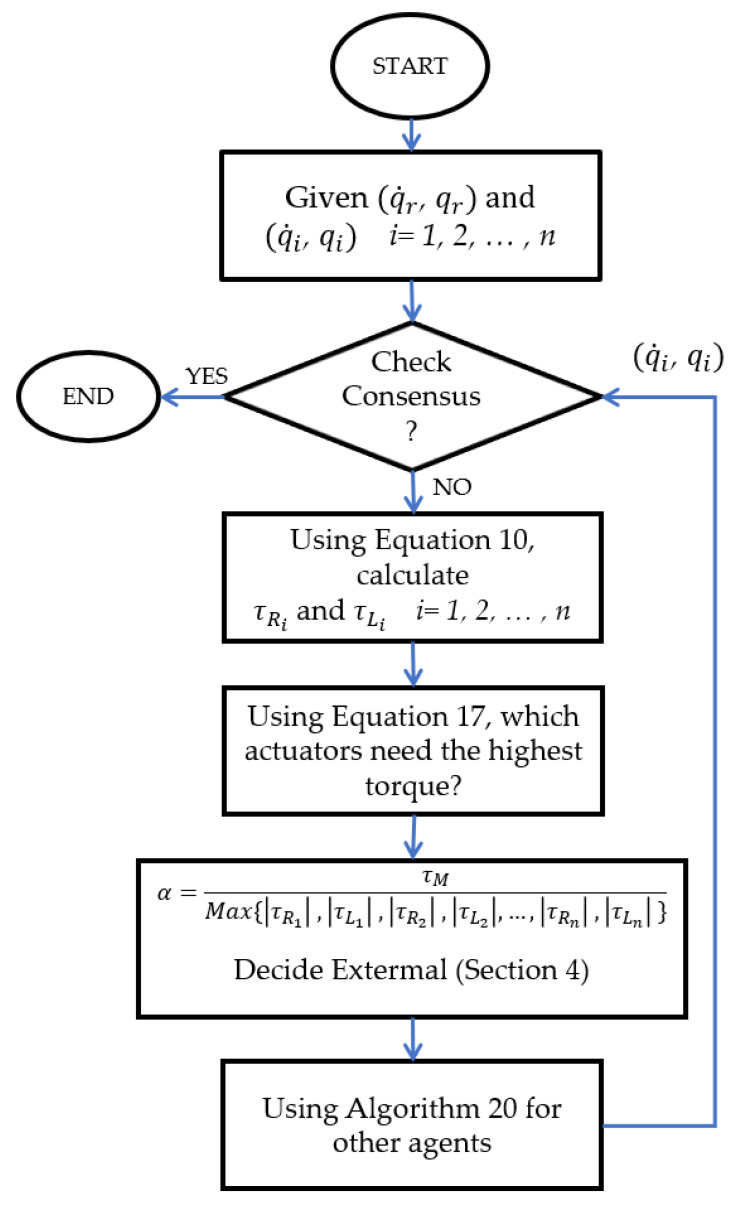
The algorithm at each time to achieve time-optimal consensus.

**Figure 4 sensors-21-07997-f004:**
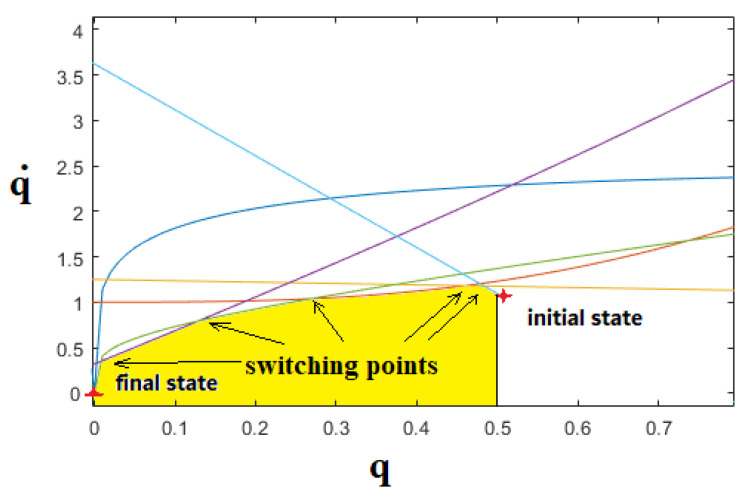
Time-optimal trajectory in the phase plane as an example. The admissible region, according to the motor constraints, is marked in a yellow colour.

**Figure 5 sensors-21-07997-f005:**
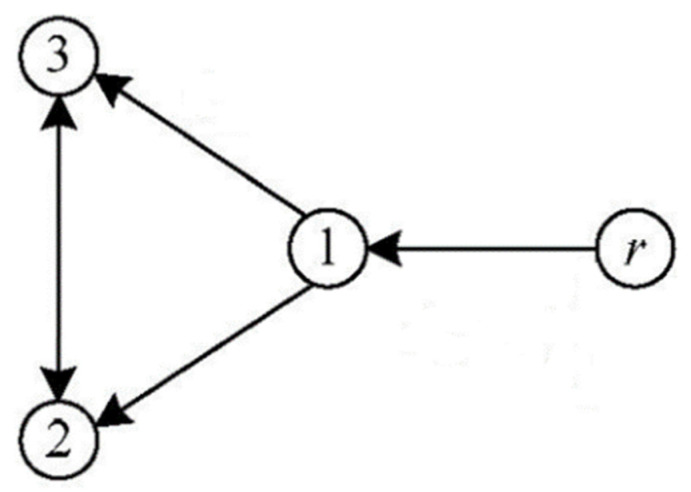
The schematic of a formation graph of the multi-robot example. The reference signal “r” is sent to “robot 1” by either the operator or a leader robot. The signal is distributed across the network based on the network topology.

**Figure 6 sensors-21-07997-f006:**
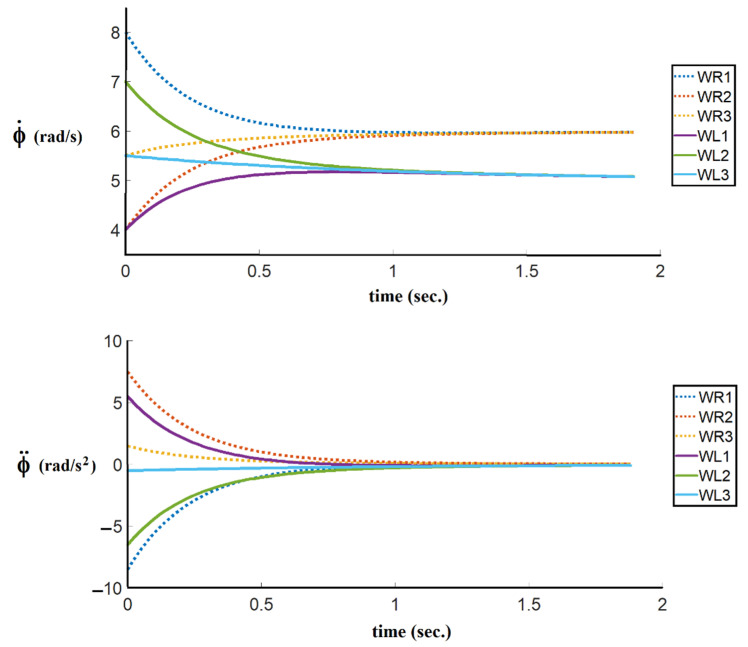
The results of the right and the left angular velocities and angular accelerations of followers without the time-optimal control coefficient.

**Figure 7 sensors-21-07997-f007:**
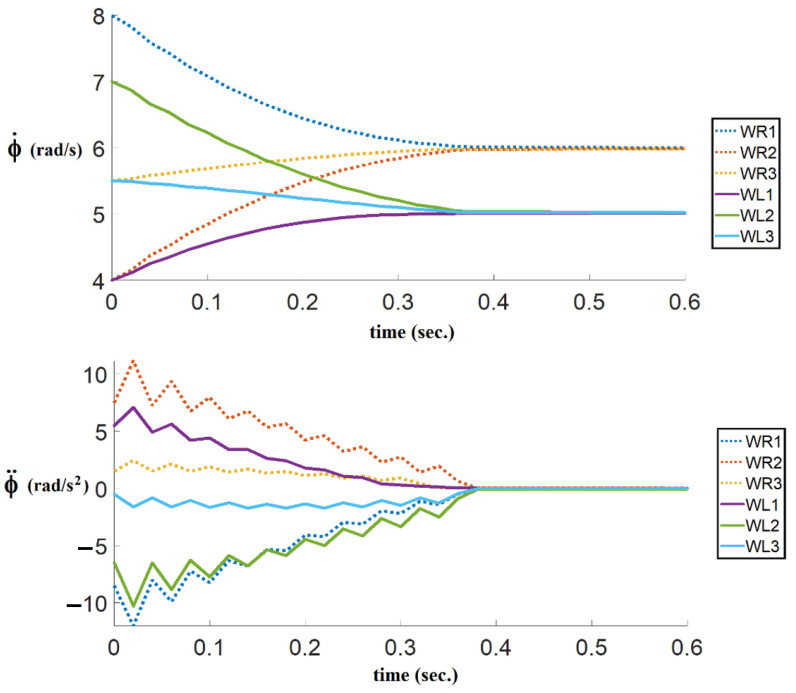
The results of the right and the left angular velocities and angular accelerations of followers with the proprosed time-optimal control consensus algorithm.

**Figure 8 sensors-21-07997-f008:**
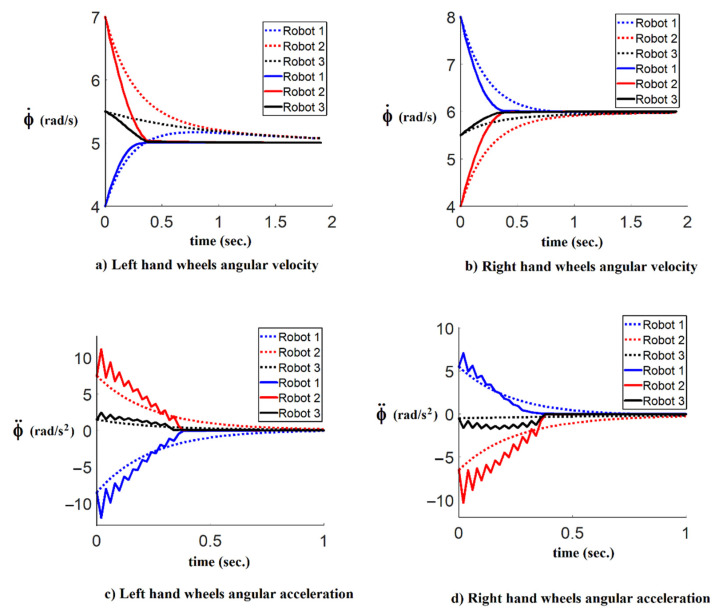
The comparison results of the right and the left angular velocities and angular accelerations of followers with and without the time-optimal control coefficient; (**a**,**b**) show the left and right hand wheels angular velocities respectively; (**c**,**d**) show the left and right angular accelerations; dashed lines show the results of the consensus algorithm, and solid lines show the results of the proposed time-optimal control consensus algorithm.

**Figure 9 sensors-21-07997-f009:**
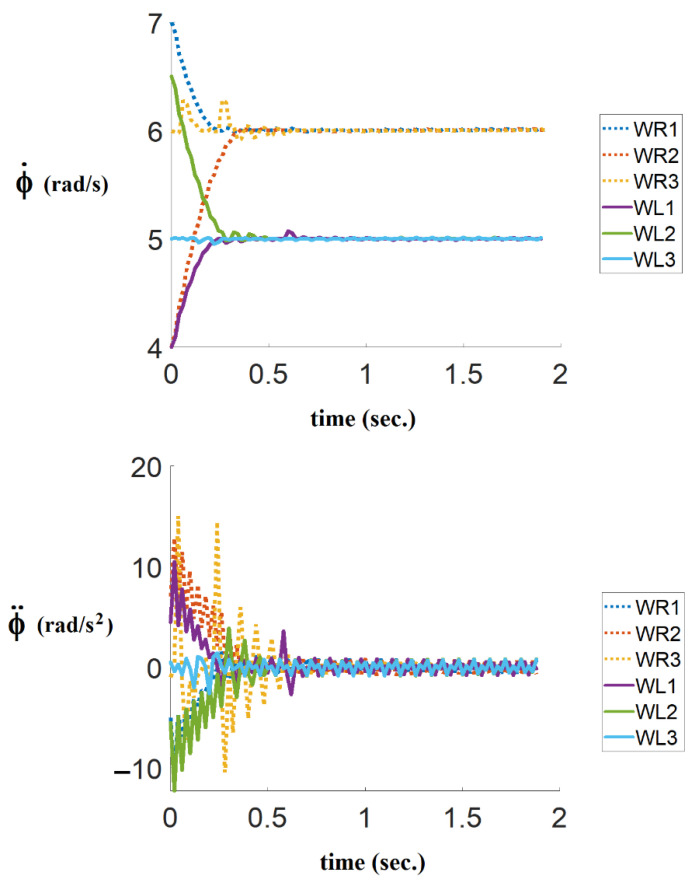
The results of the right and the left angular velocities and angular accelerations of followers with the proposed time-optimal control consensus algorithm in the presence of random noises.

**Figure 10 sensors-21-07997-f010:**
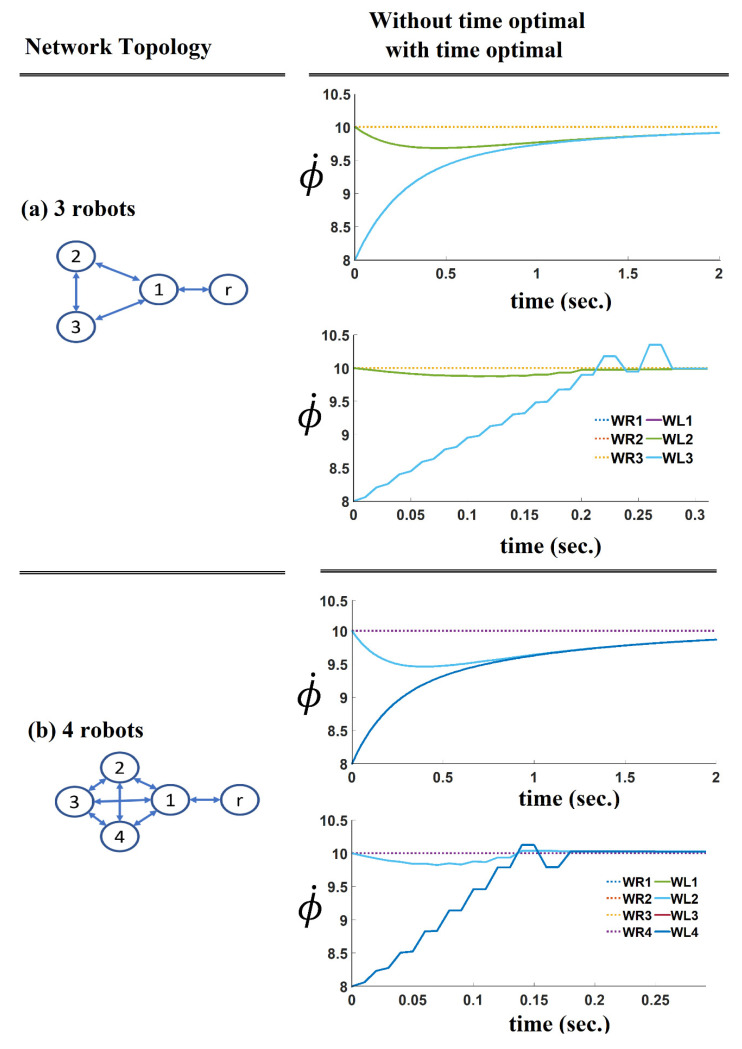
The result shows how the number of agents can influence the convergence time in the robot networks with different 5number of robots (**a**) shows a network of 3 robots; (**b**) shows a network of 4 robots; (**c**) shows a network of 5 robots; (**d**) shows a network of 6 robots.

**Figure 11 sensors-21-07997-f011:**
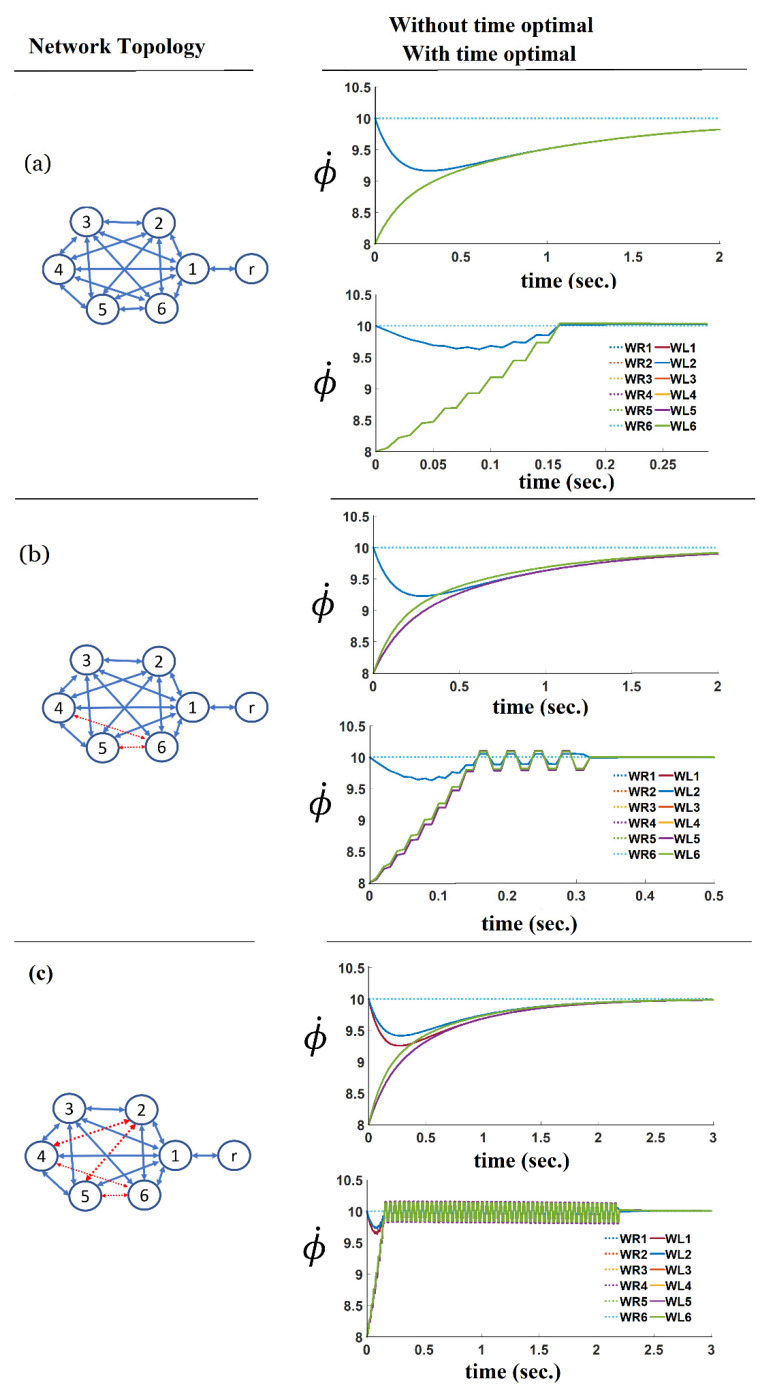
The effect of network topology on the convergence time with (**a**) a fully interconnected network; (**b**,**c**) a partially connected network with six robots.

**Table 1 sensors-21-07997-t001:** The representative works on the optimal control of multi-agents.

Ref.	Date	Proposed Method	Objectives
[21]	2014	A gradient-based optimisation algorithm, using the constraint transcription and a time scaling transform method.	An optimal parameter selection problem with continuous state inequality constraints and free terminal time.
[22]	2016	An improved gravitational search algorithm is used to optimise the trajectory of the path for multiple robots.	A multi-robot path planning problem in a dynamic environment.
[23]	2017	The direction priority sequential selection algorithm and extension-decomposition aggregation scheme are applied to solve the formation control problem and achieve collision avoidance during the formation manoeuvre.	A collision avoidance strategy based on the formation control model.
[24]	2017	Based on sliding-mode auxiliary systems, an adaptive near-optimal protocol is presented to control multi-agent systems.	A normal near-optimal protocol was designed by making an approximation of the performance index.
[25]	2017	A data-based adaptive dynamic programming method is presented using the current/past system data.	Used a discounted performance index and formulated the optimal consensus problem via the Bellman optimality principle.
[26]	2018	The fixed-time consensus theory and continuous-time zero-gradient algorithms are used	Addressed the problem of the global cost function being the sum of strictly convex local cost functions.
[28]	2019	A dynamic allocation method is proposed to increase exploration capabilities, extending them in both the inclusion phase and consensus phase of the tasks.	They solved the problems of allocation approaches that tended to trap in a local optimal and cannot obtain high-quality solutions.
[29]	2019	A constrained non-linear optimisation is combined with consensus to compute the parameters of the multi-robot formation.	A distributed method was used to solve the consensus formation of a team of aerial or mobile robots navigating with static and dynamic obstacles, when each robot has a finite communication and visibility radius.
[30]	2019	An archetypal model of distributed decision-making is used to study the capacity of the system to follow a driving signal for varying topologies and system sizes	Navigating with static and dynamic obstacles when each robot has a finite communication and visibility radius.
[31]	2020	Using the idea of CenterPoint, which is an extension of the median in higher dimensions, instead of a Tverberg partition, provides a better characterisation of the necessary and sufficient conditions guaranteeing resilient vector consensus of a multi-agent system.	Resilience guarantees improvement of the existing consensus algorithms in multi-agent networks.
[32]	2020	An alternative method to achieve a distance-based formation that used genetic algorithms to find a solution based on the distance and angle, and a constant velocity while avoiding collisions.	A parallel scheme was extended to improve the performance and find the best ways to converge to the desired distances while avoiding collisions.

## Data Availability

Not applicable.

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
