# Peer review of "Time-Optimal Velocity Tracking Control for Consensus Formation of Multiple Nonholonomic Mobile Robots"

_sensors, 2021, doi:10.3390/s21237997_

Round 1

Reviewer 1 Report

This paper considers the problem of time-optimal velocity tracking of multiple-wheeled mobile robots. The authors proposed a dedicated mathematical apparatus. They conducted simulation studies aimed at determining velocity trajectories for two cases: with and without time-optimal control. Simulation studies were performed for different numbers of robots in the network. Subsequent studies showed that full mutual communication between robots is a better solution. Presented research in my opinion is worth further development, especially comparative tests with real objects as a next step.

Author Response

Thank you for your positive feedback.

Reviewer 2 Report

I have read the whole article twice and I can only desire the authors to include experimental verification. I do understand that the purpose of paper is to present so far findings, so it is not a must. The paper is very good. review of the state of art is very good and up to date.

I vould suggest to improve figures 10 and 11 in term of bigger font sizes to it would be easier to analyse which color is which signal.

Unfortunatelly, or maybe fortunatelly i have no other sugestions for the authors since the paper is so good prepared. I'm not a native english speaker, yet everything is understandable for me.

Reviewer 3 Report

This manuscript proposed a time-optimal velocity tracking control technology for the multiple nonholonomic mobile robots. However, the innovation and advances of this work are still not well addressed in its current revision. My detailed comments are listed as below:

1、The consistency control of trajectory and velocity has also been widely studied in the field of UAVs and missiles. It is recommended to state the particularity of such problems for multiple nonholonomic robots by literatures review and technical comparison in the Introduction Section.

Ref:

Zhang Y, Tang S, Guo J. Two-stage cooperative guidance strategy using a prescribed-time optimal consensus method[J]. Aerospace Science and Technology, 2020, 100: 105641.

Yu H, Dai K, Li H, et al. Distributed cooperative guidance law for multiple missiles with input delay and topology switching[J]. Journal of the Franklin Institute, 2021.

2、As the authors said, the consistency control technology of multi-agent systems can be divided into three categories. Therefore, the authors should clearly state that which category their proposed technology belongs to. And the references in the Table 1 should focus on the same category.

3、Which parameters have influences on the convergence time? What is the specific relationship between them and the convergence time?

4、The authors did not compare their simulation performance with any previous literature. It is difficult to verify the advances of their technology.

5、How to quantitatively evaluate the anti-noise performance of the proposed control algorithm?

6、The manuscript lacks discussion on the stability of the control algorithm. For example, does it have stability under uncertain communication topology switching?

Round 2

Reviewer 3 Report

No more comments. It is now suitable for publication.

This manuscript is a resubmission of an earlier submission. The following is a list of the peer review reports and author responses from that submission.

Round 1

Reviewer 1 Report

At the beginning, I would like to congratulate the authors on a very interesting paper. The subject matter presented in the article is timely. The presented results of simulation studies are very promising, I hope that in the near future will appear the implementation of the proposed algorithm on the real object. This will allow comparing the simulation results with the measurement results. 

Author Response

Comments and Suggestions for Authors – Reviewer 1

  • At the beginning, I would like to congratulate the authors on a very interesting paper. The subject matter presented in the article is timely. The presented results of simulation studies are very promising, I hope that in the near future will appear the implementation of the proposed algorithm on the real object. This will allow comparing the simulation results with the measurement results.

Response: We would like to thank the reviewer for giving the positive feedback. As reported in the latest version of the manuscript, we extended our simulation study and examined the performance of the proposed method in different network with different condition. The latest version of the manuscript then is more comprehensive and includes the details of the extra study we have done.

Reviewer 2 Report

This work proposes a control algorithm for controlling the velocity of a group of mobile two-wheel robots, moving in formation.

Comments:

The authors claim novelty of their work by saying that "no research has been reported in which the motion time consensus along the desired formation of a group of mobile robots form and maintain a desired geometric pattern and follow the desired trajectory is addressed as the objective function."

While this might be true, your sentence is quite generic, and, most importantly, in the technical sections it does not come clear where the original contribution starts. It is appreciable that the paper provides a table with a comparison with other works, but the description of the other works' contributions is not easily comparable with your own one, i.e., it is hard to distinguish the differences.

Up to Section 3, you report equations of motions related to two-wheel robots. So the meat of your contribution starts in Section 4. I would suggest to remark, also in this section, that this is what you are proposing.

The performance evaluation is somewhat limited, you just consider a single and very simple topology, I would suggest to add more topogies, considering a larger number of nodes and shape of the formation. Additionally, you could analyse the impact of the number of nodes on the convergence time, or the impact of the transmission range (or equivalently, of the inter-node distance) on the convergence time.

As an additional question, are there any problems, from the mechanical point of view, in using the proposed alternating technique? Can this cause stress to the robots engine components?

Minor comments

In Equations (1) and (2) you use two indices, i and l. There should be a unique index i, isn't it?

row 17:

velocity conditions of multiple mobile robots system. --> velocity conditions of a multiple mobile robots system.

row 65:

the methodologies and objectives of our work are novel as it differs from the state-of-the-art. --> the methodologies and objectives of our work are novel as *they differ* from the state-of-the-art.

row 245:

algorithm used --> algorithm is used

row 260:

in the presence of noises --> in the presence of noise

Author Response

Comments and Suggestions for Authors – Reviewer 2

  • This work proposes a control algorithm for controlling the velocity of a group of mobile two-wheel robots, moving in formation. The authors claim novelty of their work by saying that "no research has been reported in which the motion time consensus along the desired formation of a group of mobile robots form and maintain a desired geometric pattern and follow the desired trajectory is addressed as the objective function." While this might be true, your sentence is quite generic, and, most importantly, in the technical sections it does not come clear where the original contribution starts. It is appreciable that the paper provides a table with a comparison with other works, but the description of the other works' contributions is not easily comparable with your own one, i.e., it is hard to distinguish the differences.

Response: We would like to sincerely thank you for your time and proving these productive comments. As suggested, we updated the text and named the specific proposed control method used in this work. The current version looks more clear and easier for the reader to follow.  

  • Up to Section 3, you report equations of motions related to two-wheel robots. So the meat of your contribution starts in Section 4. I would suggest to remark, also in this section, that this is what you are proposing.

Response: As suggested we added description in the beginning of section 4 that addresses the reviewer comment.

  • The performance evaluation is somewhat limited, you just consider a single and very simple topology, I would suggest to add more topogies, considering a larger number of nodes and shape of the formation. Additionally, you could analyse the impact of the number of nodes on the convergence time, or the impact of the transmission range (or equivalently, of the inter-node distance) on the convergence time.

Response: To address the reviewer productive comment, we extensively  studied the effect of the system performance with different network topologies as well as different numbers of robots/nodes. As the result we updated the manuscript and added sub-section 5.2 and illustrated the results in two separated figures, Figure 9 and Figure 10. The latest version looks much more comprehensive than the initial one.

  • As an additional question, are there any problems, from the mechanical point of view, in using the proposed alternating technique? Can this cause stress to the robots engine components?

Response: Yes, however as the focus of this work is on studying of the time convergence we didn’t analyse the stress to the robot component. In this work, it is assumed that the robot can afford the maximum limit torque. 

  • Minor comments: In Equations (1) and (2) you use two indices, i and l. There should be a unique index i, isn't it?

Row 17: velocity conditions of multiple mobile robots system. --> velocity conditions of a multiple mobile robots system;

Row 65: the methodologies and objectives of our work are novel as it differs from the state-of-the-art. --> the methodologies and objectives of our work are novel as *they differ* from the state-of-the-art.

Row 245: algorithm used --> algorithm is used

Row 260: in the presence of noises --> in the presence of noise

Response: we have made all the proposed corrections.

Reviewer 3 Report

The paper is superficially written and very vague. The preliminary part is presented briefly and with many ambiguities. A sketch of such a robot system would be required. It is not very clear what the authors' contribution is. It is not very clear what system the authors simulated (Figure 4). The paper seems more a juxtaposition of several parts, extracted from a research report, without making the effort to be presented as a unit. In these conditions we can only reject the work. The formal part and the template are superficially respected.

Author Response

Comments and Suggestions for Authors – Reviewer 3

  • The paper is superficially written and very vague. The preliminary part is presented briefly and with many ambiguities. A sketch of such a robot system would be required. It is not very clear what the authors' contribution is. It is not very clear what system the authors simulated (Figure 4). The paper seems more a juxtaposition of several parts, extracted from a research report, without making the effort to be presented as a unit. In these conditions we can only reject the work. The formal part and the template are superficially respected.

Response: Thanks for your comments. We tried our best to make the revised version of manuscript clearer by updating the text, adding extra sections (section 5.1 and 5.2) and also 2 new figures (Figure 9 and 10). For this purpose, we did an extensive study by extending the work and addressing the challenges in different new conditions. We did all these amendments thanks to the clear comments and suggestions of other reviewers. We are happy to address any other new scientific and clear comments in the context of this work.  

Round 2

Reviewer 2 Report

My first review round comments have been addressed by adding some text and expanding the performance evaluation.

As for the added text, this would be ok, but the authors seem to have done it in a rush, as in almost every sentence there are error. Even a section title (section 5.1) is incomplete. Avoiding this type of errors just requires an additional reading before submitting a maunuscript, so I am wondering why the authors just didn't make that very little extra effort.

I guess the added symbols (in red) in equation (1) and (2) are supposed to replace the old ones, which, by the way, are still there.

Regarding the new material provided in Section 5.2, it is ok. However, I would suggest avoid using sentences like "a surprising behaviour of the system as it is expected to see an increase of the convergence time when the number of robots increases". The relationship betweeen graph connectivity and convergence speed of consensus algorithms is a well studied topic, nothing here is surprising. I would suggest to just state that you have performed additional simulation with an increased number of nodes, considering both fully connected networks as well as networks with missing links. You might also want to say that the convergence speed of consensus algorithms in relation to the graph connectivity is a well studied problem, and outside the scope of your work.

Here below, a list of additional errors introducede while adding the new text. Please carefully proof-check your manuscript while preparing the new version.

Row 66:

of our work are novel as it they differs from the state-of-the-art. --> of our work are novel as they differ from the state-of-the-art.

Row 72:

using switching mechanism base on bang-bang control --> 

using a switching mechanism based on bang-bang control

Row 166:

time-oprtimal. --> time-optimal

Row 167:

to deal and reduce --> to reduce

Row 168:

system --> systems

Row 296:

for the different number of robots --> for a different number of robots

Row 316:

is shown --> shows

Row 318

the second trial --> a second trial

Author Response

Comments and Suggestions for Authors – Reviewer 2

My first review round comments have been addressed by adding some text and expanding the performance evaluation. As for the added text, this would be ok, but the authors seem to have done it in a rush, as in almost every sentence there are error. Even a section title (section 5.1) is incomplete. Avoiding this type of errors just requires an additional reading before submitting a manuscript, so I am wondering why the authors just didn't make that very little extra effort.

I guess the added symbols (in red) in equation (1) and (2) are supposed to replace the old ones, which, by the way, are still there.

Response: Thanks again for your time and the comments. Previously, we made all the changes and amendments suggested by you (including the equation 1 and equation 2 issues) and updated the manuscript in MS Word in the track changes style (as requested by the editor). It seems, for whatever reason, the changes were not visible on your machine. This time we uploaded three versions of the manuscript, with and without the track changes as well as the PDF file. So, we tidied up and finalised the manuscript in the “sensors-1269119-final.PDF” version for convenience.

Regarding the new material provided in Section 5.2, it is ok. However, I would suggest avoid using sentences like "a surprising behaviour of the system as it is expected to see an increase of the convergence time when the number of robots increases". The relationship between graph connectivity and convergence speed of consensus algorithms is a well studied topic, nothing here is surprising. I would suggest to just state that you have performed additional simulation with an increased number of nodes, considering both fully connected networks as well as networks with missing links. You might also want to say that the convergence speed of consensus algorithms in relation to the graph connectivity is a well studied problem, and outside the scope of your work.

Here below, a list of additional errors introduced while adding the new text. Please carefully proof-check your manuscript while preparing the new version.

Row 66:of our work are novel as it they differs from the state-of-the-art. --> of our work are novel as they differ from the state-of-the-art.

Row 72:using switching mechanism base on bang-bang control -->

using a switching mechanism based on bang-bang control

Row 166:time-oprtimal. --> time-optimal

Row 167:to deal and reduce --> to reduce

Row 168:system --> systems

Row 296:for the different number of robots --> for a different number of robots

Row 316:is shown --> shows

Row 318 the second trial --> a second trial

Response: Thanks for your comments. All the required changes have been made in the current version of the manuscript.  

Reviewer 3 Report

Dear authors,

The work has been improved but in a minor way. There are only changes in form and not in substance. The relation used in the text are elementary in this very studied field. It would have been desirable for the author to make a sketch (even elementary) of such a robot. Things can be more complex in practice than the relationships presented (3) - (7). Where are the quadratic terms in omega, which usually appear in the problems of plane dynamics ??? It may not appear, but let's see why. Section 2.1 should be completely redone and only then can an analysis of the paper be made.

There are other formal issues. For example, Section 3 does not exist or we have inappropriate expressions such as: the topology between the robots, the topology of the communication (and they are not the only ones).

Author Response

Comments and Suggestions for Authors – Reviewer 3

Dear authors, The work has been improved but in a minor way. There are only changes in form and not in substance. The relation used in the text are elementary in this very studied field. It would have been desirable for the author to make a sketch (even elementary) of such a robot. Things can be more complex in practice than the relationships presented (3) - (7). Where are the quadratic terms in omega, which usually appear in the problems of plane dynamics ??? It may not appear, but let's see why. Section 2.1 should be completely redone and only then can an analysis of the paper be made.

Response: Thanks again for your comments. The main focus of this work is the time optimal velocity tracking using bang-bang control theory in simulation level. We are aware of the control challenges in working with actual robots and in the real-world settings, however this is our next target to address. The simulation work that we presented is important before setting up any actual implementation and will help us to have a wider picture of what is going to happen when we control a network of robots. In fact, it will be difficult to design a control system for a network of robots, directly, without having the simulation study in advance as it will decrease the success rate and increase the cost. We did our best to present a comprehensive simulation work which will be helpful for those who are working in both simulation and practical implementation in the field.

There are other formal issues. For example, Section 3 does not exist or we have inappropriate expressions such as: the topology between the robots, the topology of the communication (and they are not the only ones).

Response: The errors found in naming the sections and the terms have been corrected, thanks for pointing it out.